# CESA-MCFormer: An Efficient Transformer Network for Hyperspectral Image Classification by Eliminating Redundant Information

**DOI:** 10.3390/s24041187

**Published:** 2024-02-11

**Authors:** Shukai Liu, Changqing Yin, Huijuan Zhang

**Affiliations:** School of Software, Tongji University, Shanghai 201800, China; 2131505@tongji.edu.cn (S.L.); mszhj@tongji.edu.cn (H.Z.)

**Keywords:** hyperspectral image classification, transformer, spatial attention, morphological convolution

## Abstract

Hyperspectral image (HSI) classification is a highly challenging task, particularly in fields like crop yield prediction and agricultural infrastructure detection. These applications often involve complex image types, such as soil, vegetation, water bodies, and urban structures, encompassing a variety of surface features. In HSI, the strong correlation between adjacent bands leads to redundancy in spectral information, while using image patches as the basic unit of classification causes redundancy in spatial information. To more effectively extract key information from this massive redundancy for classification, we innovatively proposed the CESA-MCFormer model, building upon the transformer architecture with the introduction of the Center Enhanced Spatial Attention (CESA) module and Morphological Convolution (MC). The CESA module combines hard coding and soft coding to provide the model with prior spatial information before the mixing of spatial features, introducing comprehensive spatial information. MC employs a series of learnable pooling operations, not only extracting key details in both spatial and spectral dimensions but also effectively merging this information. By integrating the CESA module and MC, the CESA-MCFormer model employs a “Selection–Extraction” feature processing strategy, enabling it to achieve precise classification with minimal samples, without relying on dimension reduction techniques such as PCA. To thoroughly evaluate our method, we conducted extensive experiments on the IP, UP, and Chikusei datasets, comparing our method with the latest advanced approaches. The experimental results demonstrate that the CESA-MCFormer achieved outstanding performance on all three test datasets, with Kappa coefficients of 96.38%, 98.24%, and 99.53%, respectively.

## 1. Introduction

With the continuous advancement of spectral imaging technology, hyperspectral data have achieved significant improvements in both spatial and spectral resolution. Compared to multispectral and RGB images, hyperspectral images (HSI) possess narrower bandwidths and a greater number of bands, allowing them to provide more detailed and continuous spectral information [1,2,3]. As a result, HSI have demonstrated tremendous potential in various earth observation fields [4] such as precision agriculture [5,6], urban planning [7,8], environmental management [9,10,11], and target detection [12,13,14,15]. Consequently, research on HSI classification has rapidly progressed.

While traditional HSI classification methods, such as nearest neighbor [16], Bayesian estimation [17], multinomial logistic regression [18,19], and Support Vector Machine (SVM) [20,21,22,23], have their merits in certain scenarios, these methods often have limitations in data representation and fitting capability, struggling to produce satisfactory classification results on more complex datasets. In contrast, in recent years, methods underpinned by deep learning, thanks to their outstanding feature extraction capabilities, have gradually become the focus of research in HSI classification.

Convolutional Neural Networks (CNNs) dominate the field of deep learning and are capable of accumulating in-depth spatial features through layered convolution. As such, CNNs have been extensively applied to and researched in the classification of HSI [24,25]. Notably, Roy and colleagues [26] introduced a model named HybridSN. This model initially employs a 3D-CNN to extract spatial–spectral features from spectral bands that have undergone PCA dimensionality reduction. Subsequently, it uses a 2D-CNN to delve deeper into more abstract spatial feature hierarchies. Compared to a 3D-CNN, this hybrid approach simplifies the model architecture while effectively merging spatial and spectral information. Building on this, subsequent researchers have incorporated one-dimensional convolution based on central pixels to compensate for spectral information that might be lost after PCA reduction. Examples of this approach include the Cubic-CNN model proposed by J. Wang et al. [27] and the JigsawHSI model introduced by Moraga and others [28].

The Vision Transformer (ViT) model [29], which evolved from the natural language processing (NLP) domain, has also increasingly become a focal point in the field of deep learning. The ViT model segments images into fixed-size patches and leverages embedding techniques to obtain a broader receptive field. Furthermore, with the help of multi-head attention mechanisms, it adeptly captures the dependencies between different patches, thereby achieving higher processing efficiency and remarkable image recognition performance. Consequently, numerous studies have been dedicated to exploring the application of this model in HSI classification. For instance, a research team proposed the Spatial–Spectral Transformer (SST) model in [30]. They utilized VGGNet [31], from which several convolutional layers were removed, as a feature extractor to capture spatial characteristics from hyperspectral images. Subsequently, they employed the DenseTransformer to discern relationships between spectral sequences and used a multi-layer perceptron for the final classification task. Qing et al. introduced SATNet in [32], which effectively captures spectral continuity by adding position encoding vectors and learnable embedding vectors. Meanwhile, Hong and colleagues presented the SpectralFormer (SF) model in [33]. This model adopts the Group-wise Spectral Embedding (GSE) module to encode adjacent spectra, ensuring spectral information continuity, and utilizes the Cross-layer Adaptive Fusion (CAF) technique to minimize information loss during hierarchical transmission. X. He and their team introduced the SSFTT network in [34]. This model significantly simplifies the SST structure and incorporates Gaussian-weighted feature tagging for feature transformation, thus reducing computational complexity while enhancing classification performance.In recent studies, researchers have continued to explore more lightweight and effective methods for feature fusion and extraction based on the transformer architecture. For instance, Xuming Zhang and others proposed the CLMSA and PLMSA modules [35], while Shichao Zhang and colleagues introduced the ELS2T [36].

Due to the high correlation between adjacent bands in HSI, there is a significant amount of redundant information within HSI. Commonly, to mitigate the impact of this redundancy, the methods mentioned above [26,30,31] preprocess HSI using Principal Component Analysis (PCA). However, not using PCA leads to a significant decrease in model prediction accuracy, highlighting the model’s deficiency in extracting key spectral information. As an unlearnable dimensionality reduction technique, PCA’s process is often irreversible and can lead to information loss, such as the loss of spectral continuity [37]. Models reliant on PCA may thus produce suboptimal results. In transfer learning or few-shot image classification tasks, HSI are required to feed a large number of channels into the model to preserve as much original information as possible. This input of extensive channel data elevates the demands on feature extractors, necessitating their capability to efficiently process and extract key information from these numerous channels. Moreover, different datasets might require dimensionality reduction to different extents, making the selection of appropriate dimensions for each dataset a time-consuming operation. Therefore, we propose the CESA-MCFormer, which effectively extracts key information from HSI under conditions of limited samples and numerous channels, achieving higher classification accuracy in downstream tasks without relying on PCA for dimension reduction. To achieve this, we have incorporated attention mechanisms and mathematical morphology.

Attention mechanisms have been extensively applied in various domains of machine learning and artificial intelligence. Hu et al. [38] introduced a “channel attention module” in their SE network structure to capture inter-channel dependencies. Woo et al. [39] proposed CBAM, which combines channel and spatial attention, adaptively learning weights in both dimensions to enhance the network’s expressive power and robustness. Meanwhile, Zhong et al. [40] presented a deep convolutional neural network model, integrating both a “global attention mechanism” and a “local attention mechanism” in sequence to capture both global and local contextual information. Inspired by these advancements, researchers began incorporating spatial attention into HSI classification. Several studies [41,42,43,44] combine spectral and spatial attention mechanisms, enabling adaptive selection of key features within HSI. However, in HSI classification, a common practice is to segment the HSI into small patches and classify each patch based on its center pixel. Yet, these methods do not sufficiently consider the importance of the center pixel. This approach makes the information provided by the center pixel crucial. Recent studies have recognized this, such as those cited in [45,46], which employed the Central Attention Module (CAM). This module determines feature weights by analyzing the correlation of each pixel with the center pixel. However, considering the phenomena of same material, different spectra and different materials, same spectra in HSI, relying solely on similarity to the center pixel for weight allocation might overlook important spatial information provided by other pixels. Therefore, effectively weighting the center pixel while taking global spatial information into account remains a challenge.

Mathematical Morphology (MM) primarily focuses on studying the characteristics of object morphology, processing and describing object shapes and structures using mathematical tools such as set theory, topology, and functional analysis [47]. In previous HSI classification tasks, researchers often utilized attribute profiles (APs) and extended morphological profiles (EPs) to extract spatial features more effectively [48,49,50,51]. However, this approach typically requires many structuring elements (SEs), which are non-trainable and thus unable to effectively capture dynamic feature changes. To overcome these limitations, Roy et al. proposed the Morphological Transformer (morphFormer) in [52], combining trainable MM operations with transformers, thereby enhancing the interaction between HSI features and the CLS token through learnable pooling operations. However, this method involves simultaneous dilation and erosion of spatio-spectral features, where each SE introduces a significant number of parameters. This not only risks losing fine-grained feature information during feature selection but also leads to model overfitting and reduced robustness, especially in scenarios with limited data. Hence, there is substantial room for improvement in the application of MM in HSI classification.

The core contributions of this study are as follows:We designed a flexible and efficient Center Enhanced Spatial Attention (CESA) module specifically for hyperspectral image feature extraction. This module can be easily integrated into various models, enhancing focus on areas around the center pixel while considering global spatial information;We introduced Morphological Convolution (MC) to replace the traditional linear layer feature extraction mechanism in the transformer encoder. MC selects fine-grained features through a strategy of separating and then integrating spatial and spectral features, significantly reducing the number of parameters and enhancing the model’s robustness;Utilizing these modules, we developed the CESA-MCFormer feature extractor, capable of effectively extracting key features from a multitude of channels, supporting various downstream classification tasks. We conducted in-depth ablation experiments to provide practical and theoretical insights for researchers exploring and applying similar modules.

The rest of the paper is organized as follows: In Section 2, we provide an overview of the CESA-MCFormer’s overall framework and detail our proposed CESA and MC modules. Section 3 describes the experimental datasets, results under various parameter settings, and an analysis of the model parameters. Finally, Section 4 concludes with our research findings.

## 2. Methodology

The architecture of CESA-MCFormer is illustrated in Figure 1. For an HSI patch of size c×h×w, the spectral continuity information is initially extracted through a 3D–2D Conv Block [52], and the dimensionality is transformed to 64. Subsequently, the HSI feature of size 64×h×w is fed into the Emb Block for mixing spatial and spectral features, generating a 64 × 64 feature matrix. Then, a learnable CLS token, initialized to zero, is introduced for feature aggregation, along with a learnable matrix of size 65×64, also initialized to zero, for spatial–spectral position encoding. After combining the feature map with the position encoding, it is passed through multiple iterations of the Transformer Encoder for deep feature extraction, and the extracted features are then input into the Classifier Head for downstream classification tasks. Next, we will provide a detailed introduction to the Emb Block and Transformer Encoder.

### 2.1. Emb Block

Given that the HSI patches input into the model are generally small (with a spatial size of 11×11 adopted in this study), we introduce the Emb Block to directly mix and encode global spatial features. This approach equips the model with a global receptive field before deep feature extraction, as illustrated in Figure 2. Since the information provided by the central pixel of the HSI patch is crucial, CESA is first used to weight information at different positions, aiding the model in actively eliminating redundant information. Then, we introduce a learnable weight matrix $Wa\in R^{64\times 64}$ initialized using Xavier normal initialization, composed of 64 scoring vectors. By calculating the dot product between HSI features and each scoring vector, we score the features of each pixel. The scores are then transformed into mixing weights using the softmax function. Another learnable weight matrix $Wb\in R^{64\times 64}$, initialized in the same manner, is introduced to remap the HSI features of each pixel point through matrix multiplication. Finally, by multiplying the two matrices, we mix the spatial features based on the mixing weights to obtain the final feature encoding matrix.

The overall architecture of CESA is illustrated in Figure 3. To comprehensively consider both global information and the importance of the central pixel, we meticulously designed two modules: Soft CESA and Hard CESA. Hard CESA, a non-learnable module, statically assigns higher weights to pixels closer to the center. Soft CESA, conversely, is a learnable module that uses global information as a reference, enabling the model to adaptively select more important spatial information. This design aims to effectively integrate both global and local information, enhancing the overall performance of the model.

Specifically, CESA takes an HSI or its feature map ($F_{in}$) as input. Both Hard CESA and Soft CESA calculate and output the hard probabilistic diversity map ($M_{h}$) and the soft probabilistic diversity map ($M_{s}$), respectively. The $M_{h}$ and $M_{s}$ maps are added together and then expanded along the channel dimension to match the size of $F_{in}$ before being element-wise multiplied with $F_{in}$. Finally, an optional simple convolutional module is used to adjust the dimensions of the output feature ($F_{out}$). The implementation details of both Hard CESA and Soft CESA are presented in the following sections.

#### 2.1.1. Hard CESA

The output $M_{h}$ of Hard CESA depends only on the size of $F_{in}$ and the hyperparameter *K*. For a pixel *q* in $F_{in}$, its position coordinates are defined as (*x*, *y*), and its spectral features are denoted by $p=\left[p_{1},p_{2},\ldots ,p_{c}\right]\in R^{1\times c}$. We define $q_{c}$ as the center pixel of the patch, and its coordinates in the image are defined as ($x_{c}$, $y_{c}$). The distance *d* between $q_{c}$ and *q* is defined by the following Equation (Equation 1):(1)$$\begin{array}{c} d=\max(|x-x_{c}|,|y-y_{c}|) \end{array}$$

$q_{w}$ for pixel *q* is defined as follows in Equation (Equation 2):(2)$$\begin{array}{c} q_{w}=K-\frac{d}{h}\times \left(2K-1\right) \end{array}$$
where *h* is the length of the $F_{in}$. The hyperparameter *K* (K∈[0.5,1)) controls the importance gap between the center and edge pixels. As *K* becomes larger, the weight of the center pixels becomes larger and the weight of the edge pixels becomes smaller. When K=0.5, all pixels in the patch have equal weights, and therefore Hard CESA will not have any effect.

#### 2.1.2. Soft CESA

As shown in Figure 4, Soft CESA processes $F_{in}$ into three feature maps, $F_{1}$, $F_{2}$, and $F_{3}$. $F_{1}$ and $F_{2}$ are used to represent the overall features of $F_{in}$, while $F_{3}$ is used to introduce the feature of the center pixel.

Specifically, for a pixel *q* in $F_{in}$, its position coordinates are defined as (*x*, *y*), and its spectral features are denoted by $p=\left[p_{1},p_{2},\ldots ,p_{c}\right]\in R^{1\times c}$. The value of $F_{1}$ at position (*x*, *y*), denoted as $m_{1}$(*x*, *y*), can be calculated as follows:(3)$$\begin{array}{c} m_{1}\left(x,y\right)=\max\left(p_{1},p_{2},\ldots ,p_{c}\right) \end{array}$$

The value of $F_{2}$ at position (*x*, *y*), denoted as $m_{2}$(*x*, *y*), can be calculated as follows:(4)$$\begin{array}{c} m_{2}\left(x,y\right)=\frac{1}{c}\sum_{i=1}^{c}p_{i} \end{array}$$

To effectively extract the center overall feature in Soft CESA, we introduce a central weight vector $r=\left[r_{1},r_{2},\ldots ,r_{c}\right]\in R^{c}$ to weight $F_{in}$. Therefore, the value of $F_{3}$ at position (*x*, *y*) can be represented as follows:(5)$$\begin{array}{c} m_{3}\left(x,y\right)=\frac{1}{c}\sum_{i=1}^{c}p_{i}r_{i} \end{array}$$

We extract the spectral features of the central pixel and its eight neighboring pixels, flatten them into a one-dimensional vector, and use this as the central feature vector $v_{c}\in R^{9c}$. We introduce a matrix $A_{b}\in R^{c\times (9c)}$ composed of *c* learnable spectral feature encoding vectors and a vector $l_{b}\in R^{c}$ comprised of *c* bias terms to weight and sum the spectral bands at each position. The specific formula for calculating the corresponding *r* is as follows:(6)$$\begin{array}{c} r=softmax(A_{b}v_{c}+l_{b}) \end{array}$$

Finally, we concatenate F1, F2, and F3 along the channel dimension to form the final feature matrix *F*. After passing through a convolutional layer with a kernel size of 3×3, a softmax activation function is applied to produce the final soft probabilistic diversity map $M_{s}$:(7)$$\begin{array}{c} M_{s}=sigmoid\left(\mathrm{conv}(F)\right) \end{array}$$

It can be observed that the entire CESA model uses only c×(9c+1)+(9×3+1) learnable parameters, and the parameter *c* can be flexibly adjusted through the preceding conv Block. This means that the computational cost of CESA is very low, allowing it to be easily embedded into other models without significantly increasing the complexity of the original model.

### 2.2. Transformer Encoder

The primary function of the Transformer Encoder module is to extract deep spatial–spectral features through multiple iterations. As shown in Figure 5, in each iteration, HSI features are first processed through Spectral Morph and Spatial Morph for feature selection and extraction, followed by an interaction with the CLS token through Cross Attention, aggregating the spatial–spectral features into the CLS token.

To capture multi-dimensional features, we employ a multi-head attention mechanism in Cross Attention [29]. The input CLS token and HSI features are uniformly divided into eight parts along the spectral feature dimension, each with a feature length of eight. For each segmented feature, the CLS token serves as the query $q\in R^{1\times 8}$, and the matrix formed by concatenating the CLS token and HSI features is used as the key and value $k,v\in R^{65\times 8}$. The calculation method for Cross Attention is as follows:(8)$$\begin{array}{c} X_{attn}=dropout\left(softmax\left(\frac{\left(q\times w_{q}\right)\times {\left(k\times w_{k}\right)}^{T}}{\sqrt{l}}\right)\right)\times \left(v\times w_{v}\right) \end{array}$$

In this process, $w_{q}$, $w_{k}$, and $w_{v}$$\in R^{8\times 8}$ are all learnable parameters, while *l* is the feature length, set to eight in this study. After obtaining all eight groups of $X_{attn}$, they are reassembled along the spectral feature dimension. Then, they are processed through a linear layer followed by a dropout layer, resulting in the updated $CLS\;token^{n*}\in R^{1\times 64}$. This is then added to the input $CLS\;token^{n}$ to produce the final $CLS\;token^{n+1}$.

Inspired by morphFormer [52], we have also incorporated a Spectral Morph Block and a Spatial Morph Block into our model. The overall architecture of these two modules is identical, as shown in Figure 6. Both modules process HSI features through erosion and dilation modules. After processing, the Spectral Morph block utilizes a 1×1 convolution layer (corresponding to the blue Conv block in Figure 6) to extract deeper channel information, while the Spatial Morph block uses a 3×3 convolution layer to aggregate more channel information. The Morphological Convolution (MC) we propose is represented by the erosion and dilation modules in Figure 6. Next, we will elaborate on how MC is implemented.

MC’s primary function is to eliminate redundant data during the feature extraction process, ensuring that as the depth of the encoder increases, the HSI feature retains only pivotal information.To accomplish this, we apply multiple learnable Structuring Elements (SEs) to the HS feature for morphological convolution. Through dilation, we select maximum values from adjacent features, emphasizing boundary details. In contrast, erosion allows us to identify the minimum values, effectively attenuating minor details. Additionally, directly employing SEs might inflate the parameter count, posing overfitting risks. To mitigate this, we separate the spectral and spatial SEs, significantly reducing parameters and thereby boosting the model’s resilience.

Specifically, when using SEs with a spatial size of k×k, to maintain the consistency of input and output dimensions of the module, we first reshape the spatial dimension of the HS feature into two dimensions and then pad its boundaries, resulting in the feature matrix $H\in R^{(8+(k-1))\times (8+(k-1))\times 64}$. Next, by adopting a sliding window with a stride of 1, we segment *H* into 64 sub-blocks of size k×k×64, referred to as Xpatch. Subsequently, we further decompose Xpatch in both spatial and spectral directions. Spatially, Xpatch is divided into k×k vectors of dimension 64, denoted as $\left\{X_{a1},X_{a2},\ldots ,X_{ak\times k}\right\}$. Spectrally, Xpatch is parsed into 64 vectors of dimension k×k, represented as $\left\{X_{b1},X_{b2},\ldots ,X_{b64}\right\}$. We then introduce multiple SEs groups, where each group consists of a spatial vector of length k×k and a spectral vector of length 64. For simplicity, we name one group of SEs *W*, with its spectral vector labeled $W_{a}$ and the spatial vector $W_{b}$. For any given Xpatch and *W*, the dilation operation of the morphological convolution is shown in Figure 7.

First, we add each segmented feature vector to the corresponding $W_{a}$ and $W_{b}$ at their respective positions, then take the maximum value to obtain $h_{\mathrm{dil}}\in R^{1}$:(9)$$\begin{array}{c} h_{\mathrm{dil}}\left(X_{ai},W_{a}\right)=\max_{j\in \{1,2,\ldots ,64\}}\left(X_{ai}\left(j\right)+W_{a}\left(j\right)\right) \end{array}$$
(10)$$\begin{array}{c} h_{\mathrm{dil}}\left(X_{bj},W_{b}\right)=\max_{i\in \{1,2,\ldots ,k\times k\}}\left(X_{bj}\left(i\right)+W_{b}\left(i\right)\right) \end{array}$$

Then, we introduce two learnable vectors $h_{a}\in R^{\left(k\times k\right)}$ and $h_{b}\in R^{64}$, along with two learnable bias terms $\beta_{a}$ and $\beta_{b}$. We concatenate the results from the previous step into two one-dimensional vectors, which are then dot-multiplied with $h_{a}$ and $h_{b}$, respectively, and added to $\beta_{a}$ and $\beta_{b}$, resulting in $g_{\mathrm{dil}}\in R^{1}$:(11)$$\begin{array}{c} g_{\mathrm{dil}}\left(X_{a},W_{a}\right)=\mathrm{concat}\left(h_{\mathrm{dil}}\left(X_{a1},W_{a}\right),h_{\mathrm{dil}}\left(X_{a2},W_{a}\right),\ldots ,h_{\mathrm{dil}}\left(X_{ak\times k},W_{a}\right)\right)\times h_{a}+\beta_{a} \end{array}$$
(12)$$\begin{array}{c} g_{\mathrm{dil}}\left(X_{b},W_{b}\right)=\mathrm{concat}\left(h_{\mathrm{dil}}\left(X_{b1},W_{b}\right),h_{\mathrm{dil}}\left(X_{b2},W_{b}\right),\ldots ,h_{\mathrm{dil}}\left(X_{b64},W_{b}\right)\right)\times h_{b}+\beta_{b} \end{array}$$

Finally, we concatenate the two obtained feature values to form the convolution result of that Xpatch under the specified $W_{a}$ and $W_{b}$, referred to as $f_{\mathrm{dil}}\in R^{2}$:(13)$$\begin{array}{c} f_{\mathrm{dil}}\left(X_{\mathrm{patch}},W\right)=\mathrm{concat}\left(g_{\mathrm{dil}}\left(X_{a},W_{a}\right),g_{\mathrm{dil}}\left(X_{b},W_{b}\right)\right) \end{array}$$

In actual experiments, 16 groups of *W* were used in the erosion block. Therefore, after computing all the *W* with Xpatch, the final HSI feature size obtained through the dilation module is 32×64. Similarly, for any $X_{\mathrm{patch}}$ and *W*, the following formula describes the erosion operation $f_{\mathrm{ero}}\left(X_{\mathrm{patch}},W\right)$ in the morphological convolution:(14)$$\begin{array}{cc} f_{\mathrm{ero}}\left(X_{\mathrm{patch}},W\right) & =\mathrm{concat}(g_{\mathrm{ero}}\left(X_{a},W_{a}\right),g_{\mathrm{ero}}\left(X_{b},W_{b}\right)) \end{array}$$
(15)$$\begin{array}{cc} g_{\mathrm{ero}}\left(X_{a},W_{a}\right) & =\mathrm{concat}\left(h_{\mathrm{ero}}\left(X_{a1},W_{a}\right),h_{\mathrm{ero}}\left(X_{a2},W_{a}\right),\ldots ,h_{\mathrm{ero}}\left(X_{ak\times k},W_{a}\right)\right)\times h_{a}+\beta_{a} \end{array}$$
(16)$$\begin{array}{cc} g_{\mathrm{ero}}\left(X_{b},W_{b}\right) & =\mathrm{concat}\left(h_{\mathrm{ero}}\left(X_{b1},W_{b}\right),h_{\mathrm{ero}}\left(X_{b2},W_{b}\right),\ldots ,h_{\mathrm{ero}}\left(X_{b64},W_{b}\right)\right)\times h_{b}+\beta_{b} \end{array}$$
(17)$$\begin{array}{cc} h_{\mathrm{ero}}\left(X_{ai},W_{a}\right) & =\min_{j\in \{1,2,\ldots ,64\}}\left(X_{ai}\left(j\right)-W_{a}\left(j\right)\right) \end{array}$$
(18)$$\begin{array}{cc} h_{\mathrm{ero}}\left(X_{bj},W_{b}\right) & =\min_{i\in \{1,2,\ldots ,k\times k\}}\left(X_{bj}\left(i\right)-W_{b}\left(i\right)\right) \end{array}$$

Overall, we process the 64 $X_{\mathrm{patch}}$ using 32 sets of SEs. Specifically, 16 sets are responsible for the dilation operation, while the other 16 sets handle the erosion operation. This results in two 32×64 feature matrices. After spatial–spectral separation, the required parameter count for the SEs is reduced from 2×32×k×k×64 to 2×2×16×(k×k+64+1). Additionally, MC operates similarly to traditional convolutional layers, allowing it to directly replace convolutional layers in models. This attribute endows MC with significant versatility and adaptability.

## 3. Results and Discussion

### 3.1. Dataset Description

To validate the effectiveness of the CESA-MCFormer feature extractor, we tested its performance in two types of classification tasks. Specifically, for the semantic segmentation task, we used the Indian Pines dataset (IP), Pavia University dataset (UP), and Chikusei dataset; while, for the few-shot learning (FSL) task, the datasets included the IP, UP, Chikusei dataset, Botswana dataset, KSC dataset, and Salinas Valley dataset. The detailed information about these datasets is presented in Table 1.

#### 3.1.1. Semantic Segmentation Task

In the semantic segmentation task, we randomly selected 1% of the pixels from the UP and Chikusei datasets as the training set, with the remaining pixels as the test set. Given that the Oats class in the IP dataset has only 20 pixels, we randomly extracted 5% of the pixels from the IP dataset as the training set and the rest as the test set. In constructing the training set, we did not employ any augmentation methods nor use any dimensionality reduction techniques on the datasets. The specific number of samples for each category in each dataset is shown in Table 2, Table 3 and Table 4.

#### 3.1.2. Few-Shot Learning Task

Given the extensive category requirements for few-shot learning (FSL) training, our study utilized six datasets, with the Chikusei, Botswana, KSC, and Salinas Valley used for model pretraining, and the IP and UP datasets for testing. To ensure consistency in input data sizes across all datasets in the FSL experiments, we standardized the dimensions of each dataset to 100 using BS-Nets [53].

For the datasets involved in pretraining, we selected classes with over 250 samples, randomly allocating 50 samples to the support set and 200 to the query set for each class. Specifically, Chikusei contributed 17 classes, Botswana 8, KSC 9, and Salinas Valley 16, totaling 50 distinct training classes. After pretraining, we randomly selected 10 samples from each class in the IP and UP datasets for model fine-tuning and testing.

In our study, during the pretraining on the IP dataset, each task randomly selected 16 out of 50 available classes, following a 16-way, 10-shot method. For the UP dataset, each training iteration randomly chose nine classes, also using a 10-shot approach. For each class, the support set consisted of 10 randomly selected samples out of 50, while the query set used all 200 samples. Moreover, no form of data augmentation was used to expand the datasets, neither in the pretraining nor in the fine-tuning stages.

### 3.2. Training Details and Evaluation Indicators

#### 3.2.1. Configuration

All experiments were designed and conducted using PyTorch on a Ubuntu 18.04 x64 machine with 13th Gen Intel(R) Core(TM) i5-13600KF CPU, 32GB RAM, and an NVIDIA Geforce RTX 4080 16GB GPU.

#### 3.2.2. Training Details

In our semantic segmentation task, we directly classify using the cls token connected to a fully connected layer, as depicted in the Classifier Head block in Figure 1. The Adam optimizer is used with a learning rate of 0.001, and CrossEntropy Loss functions as the loss criterion. For models such as HybridSN [26], Vit [29], SF [33], and SSFTT [34], we maintain a batch size of 64. For morphFormer [52] and our developed CESA-MCFormer, the batch size is set at 32. Table 5 displays the floating point operations (FLOPS) and the number of parameters for various models.

For the FSL task, we incorporate two sets of trainable weights, summing HSI features weighted along spatial and spectral directions to create two feature vectors of length 64 each. These are concatenated with the cls token, forming a final vector of length 192. We average features of 10 samples from each class in the support set to represent class prototypes, with classification based on distances between query set features and these prototypes. For the convolutional network (HybridSN), we alter the final linear layer’s output to a 192-length feature vector for uniformity in feature output length across models. The SGD optimizer is employed, setting the learning rate at 0.00001 and weight decay at 0.0005.

#### 3.2.3. Evaluation Indicators

We used four quantitative evaluation metrics including overall accuracy (OA), average accuracy (AA), kappa coefficient (κ), and class-specific accuracy to quantitatively analyze the effectiveness of CESA. The higher the values of these metrics, the better the classification performance.

### 3.3. Semantic Segmentation Task Experimental Results

#### 3.3.1. Classification Results

To verify the advanced nature of our CESA-MCFormer model, we conducted comparative experiments with several recently proposed models. These include HybridSN [26], Vit [29], SF [33], and SSFTT [34], which originally required PCA for dimensionality reduction in their respective papers. Conversely, morphFormer [52] does not require such reduction. In our experiments, we used a patch size of 11×11 as the input for all models, with *K* set to 0.8 in Hard CESA.

Table 6, Table 7 and Table 8 display the classification results of various models without using PCA dimensionality reduction, while Figure 8 and Figure 9 present the visualization results on the IP and UP datasets. The experimental data demonstrate that CESA-MCFormer exhibits superior performance across all datasets, highlighting its exceptional feature extraction capability in the presence of abundant redundant information. Moreover, as observed from Figure 8 and Figure 9, the combination of EMB Block and MC to eliminate redundancy has notably enhanced the model’s accuracy in classifying complex pixels, especially in edge areas and categories with limited samples. This further underscores the outstanding performance of CESA-MCFormer.

Table 9 and Table 10 present the classification results of HybridSN, SpectralFormer, and SSFTT on the IP and UP datasets after applying PCA dimensionality reduction. Specifically, the dimensionality of the IP dataset was reduced to 30, while that of the UP dataset was reduced to 15. The results indicate that the performance of HybridSN, SpectralFormer, and SSFTT improved significantly after PCA reduction. However, their accuracy still falls short of our CESA-MCFormer model.

#### 3.3.2. Ablation Experiment

To better understand the roles of CESA and MC within the model, we conducted several ablation studies on the IP dataset. In these experiments, we continued to adopt the hyperparameter settings from the previous section. Given the similar overall architecture of CESA-MCFormer and morphFormer, and the outstanding performance of morphFormer when compared to other models, we chose morphFormer as our baseline and built upon it by adding modules.

Table 11 demonstrates the influence of MC and CESA on the final classification performance of the model. It is evident from the table that both modules have significantly enhanced the OA. Furthermore, when both modules are used in conjunction, there is an additional improvement in accuracy. However, in contrast to the pronounced improvement in AA brought about by CESA, the contribution of MC is relatively limited. This can be primarily attributed to the lower classification accuracy for classes with fewer samples. In scenarios with limited sample sizes, prior information becomes particularly crucial. Without the spatial prior information provided by CESA, relying solely on MC to process hyperspectral features that encapsulate comprehensive spatial information proves to be more challenging.

To further validate the superiority of CESA, we replaced it with the traditional Spatial Attention block (SA) and CAM in the CESA-MCFormer model and conducted comparative experiments. The results, presented in Table 12, demonstrate that CESA achieved the highest classification accuracy. This is attributed to CESA’s combination of Hard and Soft components, which not only incorporate prior information but also ensure the learnability of the entire module. Thus, CESA effectively amalgamates the advantages of SA and CAM, leading to improved performance.

### 3.4. Few-Shot Learning Task Experimental Results

To assess the generality and effectiveness of CESA-MCFormer with extremely limited samples, we conducted FSL experiments on the IP and UP datasets, using only 10 samples per class. The experimental results, as shown in Table 13 and Table 14, indicate that CESA-MCFormer achieved optimal performance on both datasets.

## 4. Conclusions

This paper presents the CESA-MCFormer feature extractor, which boosts the model’s feature extraction capabilities with a “selection–extraction” strategy, enabling effective image feature extraction without reliance on PCA. CESA enables the model to incorporate spatial prior knowledge guided by an attention mechanism while maintaining the learnability of the module. The MC module introduces learnable pooling operations that effectively filter key information during deep feature extraction. Additionally, CESA-MCFormer adapts to various classification tasks by modifying the classifier head, and both CESA and MC modules can be flexibly integrated into other models to improve their feature extraction performance. Comparisons with other models in semantic segmentation and FSL tasks confirm the versatility and effectiveness of CESA-MCFormer, and ablation studies of CESA and MC attest to the efficacy of these two components.

The CESA-MCFormer has demonstrated exceptional versatility in surface object classification tasks. In future research, we intend to further explore the model’s application to subsurface exploration tasks (such as soil composition analysis) and are committed to further optimizing its performance.

## Figures and Tables

**Figure 1 sensors-24-01187-f001:**
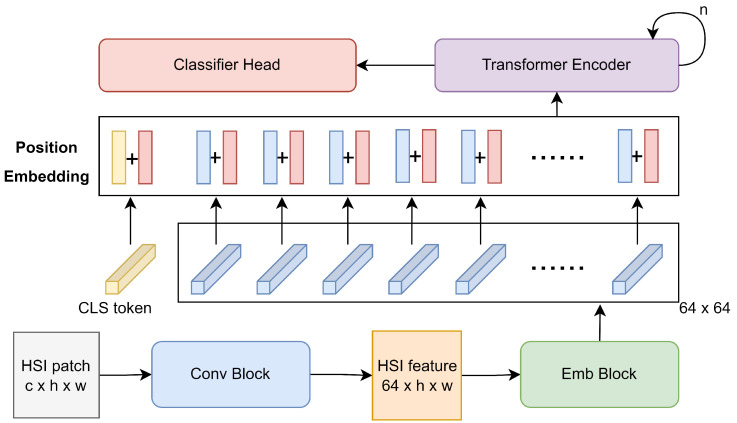
Overall architecture of CESA-MCFormer.

**Figure 2 sensors-24-01187-f002:**
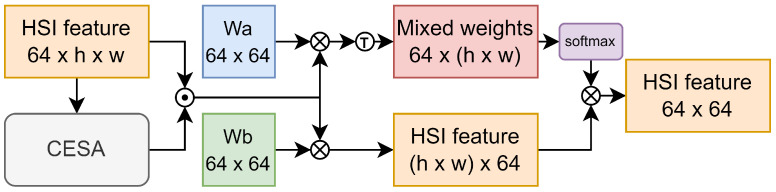
Overall architecture of EMB Block.The symbol “T” represents the transpose of a matrix. The symbol “·” represents element-wise multiplication of matrices, and the symbol ”x” denotes matrix multiplication.

**Figure 3 sensors-24-01187-f003:**
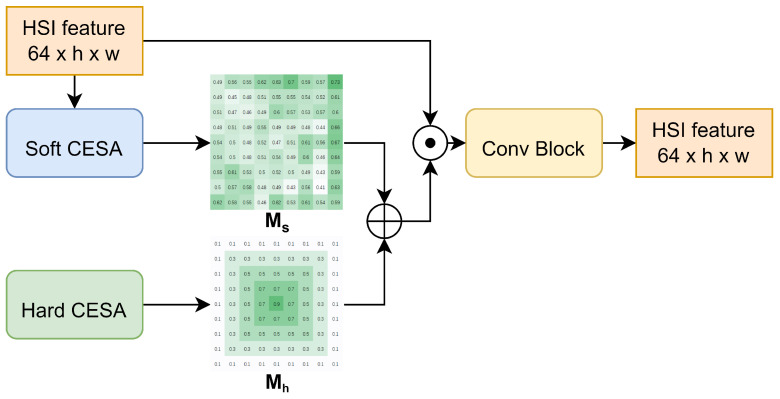
Overall architecture of CESA.

**Figure 4 sensors-24-01187-f004:**
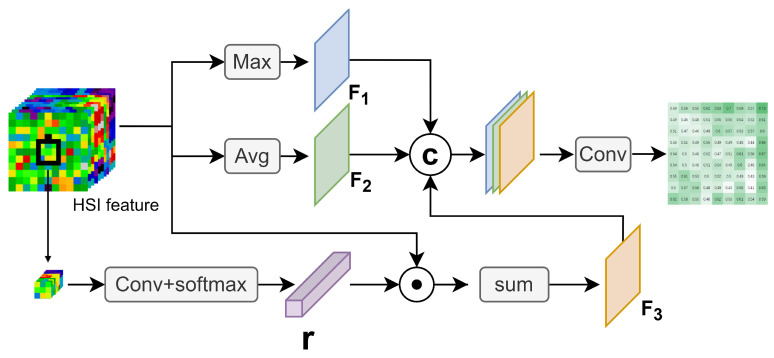
Overall architecture of Soft CESA.

**Figure 5 sensors-24-01187-f005:**
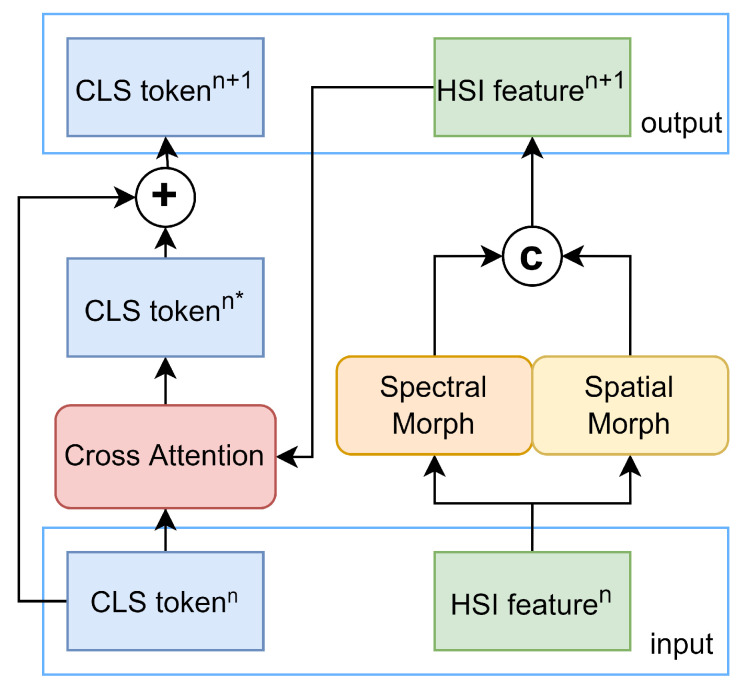
Overall architecture of Transformer Encoder.

**Figure 6 sensors-24-01187-f006:**
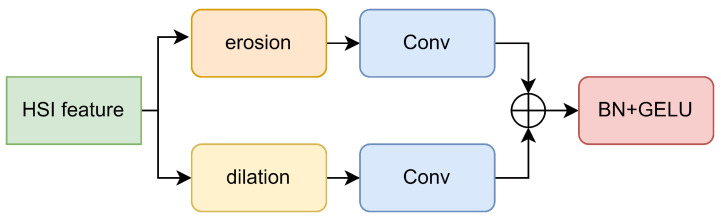
Overall architecture of the Spectral Morph Block and Spatial Morph Block.

**Figure 7 sensors-24-01187-f007:**
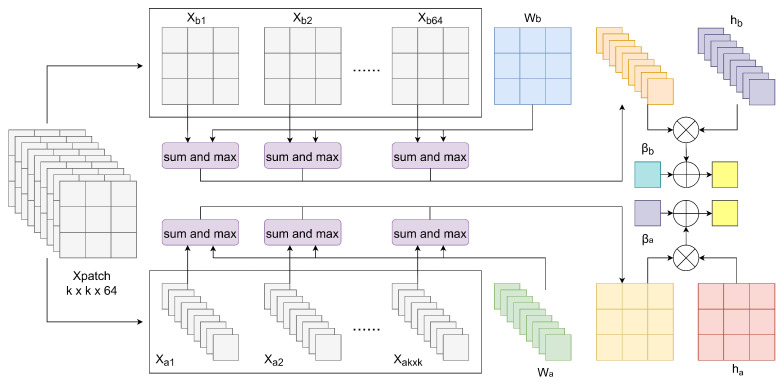
Overall architecture of dilation block.

**Figure 8 sensors-24-01187-f008:**
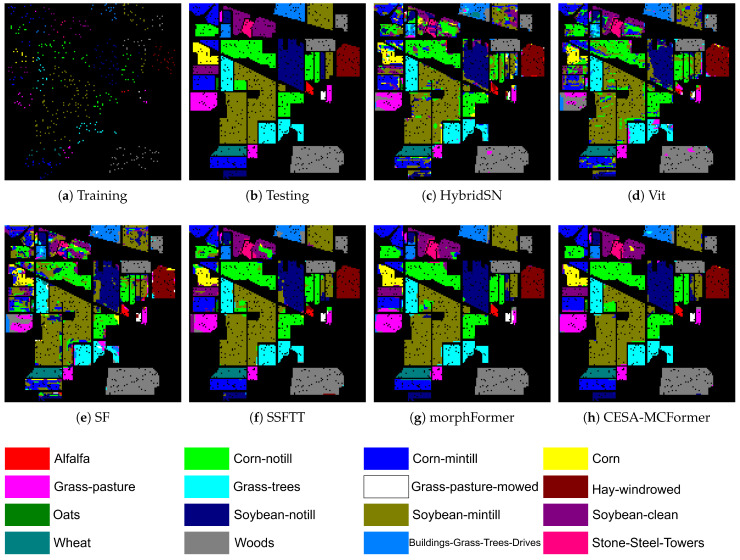
Visualization results on the IP dataset.

**Figure 9 sensors-24-01187-f009:**
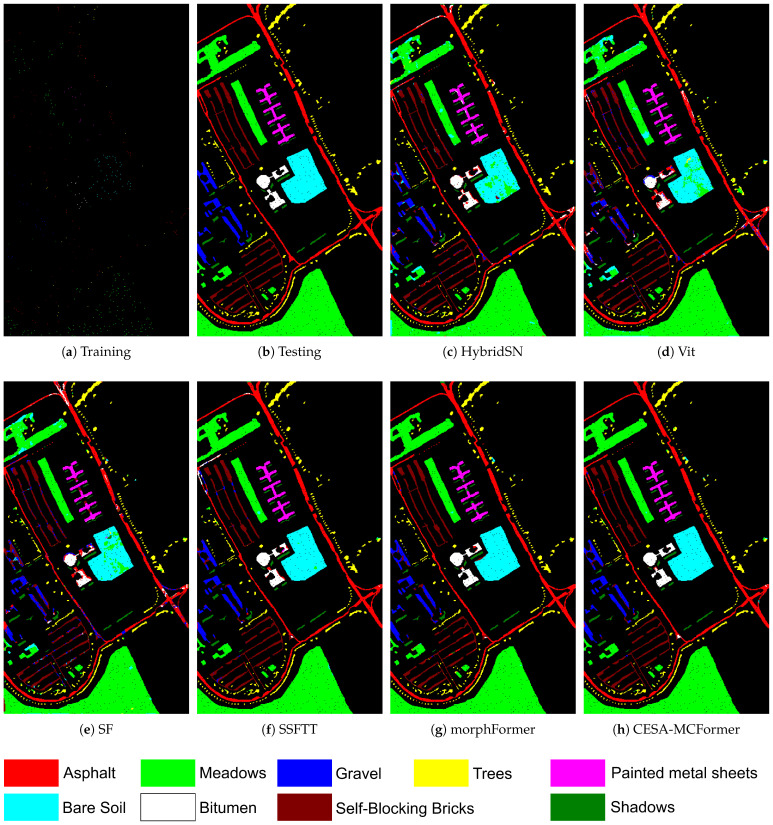
Visualization results on the UP dataset.

**Table 1 sensors-24-01187-t001:** Dataset Information.

Dataset	Image Size	Number of Classes	Number of Bands
IP	145 × 145	16	220
UP	610 × 340	9	103
Chikusei	2517 × 2335	19	128
Botswana	1476 × 256	14	145
KSC	512 × 614	13	176
Salinas Valley	512 × 217	16	224

**Table 2 sensors-24-01187-t002:** Detailed information on the training and testing data samples for each class in the IP dataset.

Class No.	Class Name	Training	Testing
1	Alfalfa	3	43
2	Corn-notill	72	1356
3	Corn-mintill	42	788
4	Corn	12	225
5	Grass-pasture	25	458
6	Grass-trees	37	693
7	Grass-pasture-mowed	2	26
8	Hay-windrowed	24	454
9	Oats	1	19
10	Soybean-notill	49	923
11	Soybean-mintill	123	2332
12	Soybean-clean	30	563
13	Wheat	11	194
14	Woods	64	1201
15	Buildings-Grass-Trees-Drives	20	366
16	Stone-Steel-Towers	5	88
	Total	520	9729

**Table 3 sensors-24-01187-t003:** Detailed information on the training and testing data samples for each class in the UP dataset.

Class No.	Class Name	Training	Testing
1	Asphalt	67	6564
2	Meadows	187	18,462
3	Gravel	21	2078
4	Trees	31	3033
5	Painted metal sheets	14	1331
6	Bare Soil	51	4978
7	Bitumen	14	1316
8	Self-Blocking Bricks	37	3645
9	Shadows	10	937
	Total	432	42,344

**Table 4 sensors-24-01187-t004:** Detailed information on the training and testing data samples for each class in the Chikusei dataset.

Class No.	Class Name	Training	Testing
1	Water	29	2816
2	Bare soil (school)	29	2830
3	Bare soil (park)	3	283
4	Bare soil (farmland)	49	4830
5	Natural plants	43	4254
6	Weeds in farmland	12	1096
7	Forest	206	20,310
8	Grass	66	6449
9	Rice field (grown)	134	13,235
10	Rice field (first stage)	13	1255
11	Row crops	60	5901
12	Plastic house	22	2171
13	Manmade (non-dark)	13	1207
14	Manmade (dark)	77	7587
15	Manmade (blue)	5	426
16	Manmade (red)	3	219
17	Manmade grass	11	1029
18	Asphalt	9	792
19	Paved ground	2	143
	Total	786	76,806

**Table 5 sensors-24-01187-t005:** The floating point operations (FLOPS) and the number of parameters for various models. “CESA-MCFormer *” refers to the CESA-MCFormer model without the inclusion of the CESA.

Class No	HybridSN	Vit	SF	SSFTT	morphFormer	CESA-MCFormer	CESA-MCFormer *
FLOPs (M)	44.80	75.34	24.91	29.88	41.78	32.24	27.67
Params (M)	1.12	0.61	0.11	0.51	0.25	0.36	0.25

**Table 6 sensors-24-01187-t006:** Classification accuracy of various models on the IP dataset (without PCA).

Class No	HybridSN	Vit	SF	SSFTT	morphFormer	CESA-MCFormer
1	62.79	39.53	83.72	58.14	**97.67**	93.02
2	73.45	74.78	64.16	91.81	92.92	**96.61**
3	70.05	69.29	50.25	94.54	88.07	**95.81**
4	67.56	57.33	53.78	77.78	**91.11**	90.22
5	76.86	46.51	57.42	79.91	89.74	**95.85**
6	94.95	86.29	81.53	98.70	99.42	**100.00**
7	46.15	42.31	76.92	**100.00**	**100.00**	**100.00**
8	96.70	95.15	92.51	**100.00**	99.34	99.34
9	63.16	26.32	73.68	**89.47**	31.58	84.21
10	76.60	73.67	76.71	87.22	**95.56**	95.12
11	81.86	84.95	83.40	97.51	97.13	**97.64**
12	44.76	51.33	45.83	82.77	88.28	**95.38**
13	**100.00**	**100.00**	98.97	**100.00**	97.94	**100.00**
14	98.17	96.92	99.25	98.83	**99.75**	99.17
15	59.84	73.50	84.43	85.79	**91.53**	87.98
16	55.68	**95.45**	51.14	97.73	**100.00**	**100.00**
OA (%)	79.24	78.38	75.59	93.15	94.96	**96.82**
AA (%)	73.04	69.58	73.36	90.01	91.25	**95.65**
κ (%)	76.28	75.17	71.96	92.17	94.26	**96.38**

**Table 7 sensors-24-01187-t007:** Classification accuracy of various models on the UP dataset (without PCA).

Class No	HybridSN	Vit	SF	SSFTT	morphFormer	CESA-MCFormer
1	89.98	91.97	83.32	94.70	98.00	**98.35**
2	95.54	93.90	95.38	99.70	99.69	**99.92**
3	75.07	47.50	59.58	87.73	87.05	**89.56**
4	94.16	92.81	89.48	96.24	**96.74**	95.12
5	96.39	99.85	89.41	99.92	93.09	**100.00**
6	84.39	84.23	85.01	99.38	99.08	**100.00**
7	82.14	62.54	63.91	90.20	97.04	**98.33**
8	92.76	91.39	87.46	95.80	95.69	**99.18**
9	98.40	99.79	**99.89**	98.51	97.87	97.55
OA (%)	91.70	89.23	88.36	97.40	97.85	**98.67**
AA (%)	89.87	84.89	83.72	95.80	96.27	**97.56**
κ (%)	89.00	85.75	84.59	96.56	97.15	**98.24**

**Table 8 sensors-24-01187-t008:** Classification accuracy of various models on the Chikusei dataset (without PCA).

Class No	HybridSN	Vit	SF	SSFTT	morphFormer	CESA-MCFormer
1	97.41	94.74	91.58	99.61	**99.96**	99.15
2	95.23	94.03	99.01	95.30	**99.82**	99.72
3	0.00	18.37	23.67	46.64	22.61	**85.16**
4	99.79	97.67	98.44	97.48	98.69	**99.88**
5	99.79	96.94	97.32	99.98	**100.00**	99.98
6	97.90	97.26	93.89	91.88	**98.54**	91.79
7	**100.00**	**100.00**	**100.00**	**100.00**	**100.00**	**100.00**
8	99.74	98.71	99.18	**100.00**	**100.00**	99.94
9	99.81	99.89	99.99	**100.00**	99.77	**100.00**
10	97.37	99.20	95.30	**100.00**	**100.00**	**100.00**
11	**100.00**	99.75	**100.00**	99.83	99.76	**100.00**
12	96.96	93.74	**98.11**	96.32	97.14	96.78
13	95.53	95.53	94.86	95.53	95.53	**95.53**
14	99.79	98.87	99.00	99.97	99.91	**100.00**
15	99.06	93.19	95.54	**100.00**	99.30	99.53
16	**100.00**	98.17	99.54	93.61	**100.00**	**100.00**
17	99.03	86.10	97.57	**100.00**	**100.00**	**100.00**
18	79.80	91.41	88.76	90.03	97.85	**99.37**
19	16.78	31.47	42.66	88.11	88.11	**95.10**
OA (%)	98.65	97.97	98.38	99.01	99.34	**99.59**
AA (%)	88.10	88.69	90.23	94.44	94.58	**98.00**
κ (%)	98.44	97.65	98.13	98.85	99.24	**99.53**

**Table 9 sensors-24-01187-t009:** Classification accuracy of various models on the IP dataset (with PCA). “CESA-MCFormer *” refers to the CESA-MCFormer model without PCA.

Class No	HybridSN	SF	SSFTT	CESA-MCFormer	CESA-MCFormer *
1	72.09	51.16	**100.00**	76.74	93.02
2	92.99	78.17	94.91	96.02	**96.61**
3	**98.98**	73.86	98.35	95.18	95.81
4	94.67	61.78	**100.00**	88.89	90.22
5	90.83	84.50	**98.69**	90.17	95.85
6	99.86	93.94	99.13	99.42	**100.00**
7	57.69	11.54	73.08	**100.00**	**100.00**
8	94.49	**100.00**	99.56	**100.00**	99.34
9	68.42	10.53	52.63	**100.00**	84.21
10	91.12	85.59	**97.29**	94.58	95.12
11	96.57	87.78	97.60	**98.37**	97.64
12	86.50	69.45	84.37	88.28	**95.38**
13	97.94	98.45	**100.00**	**100.00**	**100.00**
14	98.92	90.67	99.67	**99.92**	99.17
15	**96.45**	62.02	93.44	90.16	87.98
16	48.86	81.82	95.45	**100.00**	**100.00**
OA (%)	94.60	83.33	96.78	96.23	**96.82**
AA (%)	86.65	71.33	92.76	94.86	**95.65**
κ (%)	93.84	80.97	96.33	95.70	**96.38**

**Table 10 sensors-24-01187-t010:** Classification accuracy of various models on the UP dataset (with PCA). “CESA-MCFormer *” refers to the CESA-MCFormer model without PCA.

Class No	HybridSN	SF	SSFTT	CESA-MCFormer	CESA-MCFormer *
1	**98.86**	88.82	98.28	98.49	98.35
2	99.96	96.59	**99.97**	99.87	99.92
3	85.51	71.80	90.38	**91.77**	89.56
4	92.91	91.16	**97.36**	95.19	95.12
5	**100.00**	95.87	**100.00**	**100.00**	**100.00**
6	99.76	79.23	98.49	99.16	**100.00**
7	97.04	75.08	**98.94**	98.02	98.33
8	92.76	84.94	91.44	98.30	**99.18**
9	93.17	92.32	93.06	95.73	**97.55**
OA (%)	97.69	89.95	97.96	98.56	**98.67**
AA (%)	95.55	86.20	96.44	97.39	**97.56**
κ (%)	96.93	86.58	97.29	98.09	**98.24**

**Table 11 sensors-24-01187-t011:** Ablation study results for CESA and MC. “CESA” stands for replacing the Emb Block, and “MC” stands for replacing the Transformer Encoder.

CESA	MC+	OA (%)	AA (%)	κ (%)
		94.96	91.25	94.26
√		95.86	93.06	95.27
	√	95.80	91.50	95.21
√	√	**96.82**	**95.65**	**96.38**

**Table 12 sensors-24-01187-t012:** Comparative experiments of the traditional Spatial Attention block (SA), CAM, and CESA.

	OA (%)	AA (%)	κ (%)
SA	95.79	91.76	95.19
CAM	95.94	93.17	95.37
CESA	**96.82**	**95.65**	**96.38**

**Table 13 sensors-24-01187-t013:** Classification accuracy of various models on the IP dataset.

	HybridSN	SSFTT	morphForm	CESA-MCFormer
OA (%)	68.90	69.56	71.26	**73.29**
AA (%)	81.10	81.87	82.97	**84.61**
κ (%)	65.17	65.85	67.82	**70.05**

**Table 14 sensors-24-01187-t014:** Classification accuracy of various models on the UP dataset.

	HybridSN	SSFTT	morphForm	CESA-MCFormer
OA (%)	70.81	74.43	74.03	**77.54**
AA (%)	77.90	79.39	78.90	**83.55**
κ (%)	63.44	67.17	66.71	**71.46**

## Data Availability

Data are contained within the article.

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
