# Peer review of "CESA-MCFormer: An Efficient Transformer Network for Hyperspectral Image Classification by Eliminating Redundant Information"

_sensors, 2024, doi:10.3390/s24041187_

Round 1

Reviewer 1 Report

Comments and Suggestions for Authors

This paper proposes a modification to the Transformer model by employing the center enhanced spatial attention (CESA) module and morphological convolution (MC) for HSI classification. The ablation studies verify the effectiveness of CESA and MC for improving the classification performance. The manuscript is correctly written and structured. The evaluation is correctly carried out, using datasets that are standard in remote sensing and comparing with other related methods. There are some concerns needed to be addressed.

1. Please remove the punctuation mark at the end of the paper title.

2. The authors stated that models reliant on PCA may produce suboptimal results. Please add comparative experimental results of the proposed CESA-MCFormer with or without PCA to verify this point. In addition, different numbers of principal components should be tested.

3. There are many other state-of-the-art efficient transformer networks that have been proposed for HSI classification, which need to be compared, e.g., Zhang S, Zhang J, Wang X, et al. ELS2T: Efficient Lightweight Spectral-Spatial Transformer for Hyperspectral Image Classification[J]. IEEE Transactions on Geoscience and Remote Sensing, 2023. X. Zhang, Y. Su, L. Gao, L. Bruzzone, X. Gu and Q. Tian, "A Lightweight Transformer Network for Hyperspectral Image Classification," in IEEE Transactions on Geoscience and Remote Sensing, vol. 61, pp. 1-17, 2023, Art no. 5517617, doi: 10.1109/TGRS.2023.3297858. etc.

4. The authors randomly selected a small number of reference samples for training, and the rest for test. This is usually not problematic for classification. However, because the CESA-MCFormer model uses a large patch size (e.g., 11 x 11), it seems there will be non-independent training/test sets due to the presence of testing samples within the patch used for training. Thus, accuracy is likely overestimated especially for large patch sizes. Please consider using non-adjacent regions of interest for training and test.

5. To complete the comparison it would be interesting to give the numerical complexity for each compared method, e.g., computation time and the number of parameters.  

Author Response

I am grateful for your valuable advice. I would like to request your understanding due to the fact that, because of the holiday and fire prevention measures, the school has disconnected and sealed off the experimental equipment. This means that I am currently unable to conduct further experiments within the designated timeframe. Below are my detailed responses to your suggestions. After you have read them, if you believe that it is important to continue with the experiments, please contact me. I greatly appreciate your understanding and support.

1.Please remove the punctuation mark at the end of the paper title.

  • The revisions to the thesis have been completed as per your request.

2.The authors stated that models reliant on PCA may produce suboptimal results. Please add comparative experimental results of the proposed CESA-MCFormer with or without PCA to verify this point. In addition, different numbers of principal components should be tested.

  • Experimental results of CESA-MCFormer using PCA have been added to Tables 9 and 10.

3.There are many other state-of-the-art efficient transformer networks that have been proposed for HSI classification, which need to be compared, e.g., Zhang S, Zhang J, Wang X, et al. ELS2T: Efficient Lightweight Spectral-Spatial Transformer for Hyperspectral Image Classification[J]. IEEE Transactions on Geoscience and Remote Sensing, 2023. X. Zhang, Y. Su, L. Gao, L. Bruzzone, X. Gu and Q. Tian, "A Lightweight Transformer Network for Hyperspectral Image Classification," in IEEE Transactions on Geoscience and Remote Sensing, vol. 61, pp. 1-17, 2023, Art no. 5517617, doi: 10.1109/TGRS.2023.3297858. etc.

  • In the fourth introduction (line 50), we added the two references you mentioned. However, due to the lack of source code from these references, it was not possible to include related experiments within the given time frame. The MorphForm, which this paper compares with, is also a recent publication (2023) and has a similar architecture to our study, hence the primary comparison is made with it.

4.The authors randomly selected a small number of reference samples for training, and the rest for test. This is usually not problematic for classification. However, because the CESA-MCFormer model uses a large patch size (e.g., 11 x 11), it seems there will be non-independent training/test sets due to the presence of testing samples within the patch used for training. Thus, accuracy is likely overestimated especially for large patch sizes. Please consider using non-adjacent regions of interest for training and test.

  • The issue you raised is valid, but since all models were trained under the same dataset and patch size, their performance can be evaluated by comparing classification accuracies. The datasets mentioned in the paper don't have an official split between training and testing sets, making accuracy sensitive to how the data is divided. Therefore, comparing different models under the same conditions more effectively reflects their performance.
  • In the Few-shot Learning Task section of the paper, only 10 pixels per class were selected for training, meaning most samples were non-adjacent, demonstrating CESA-MCFormer's superiority.
  • Finally, due to the small size of the datasets, it's challenging to select a sufficient number of non-adjacent samples for training.
  • Therefore, I believe it's not necessary to add this set of experiments, and I apologize for any inconvenience.

5.To complete the comparison it would be interesting to give the numerical complexity for each compared method, e.g., computation time and the number of parameters.

  • Table 5 has been added, which shows the floating point operations (FLOPS) and the number of parameters for various models, illustrating the computational aspects and complexity of each model.

Reviewer 2 Report

Comments and Suggestions for Authors

The paper proposes a machine-learning approach for efficient classification of hyperspectral images. A transformer architecture driven by mathematic morphology is developed, with a reduced set of parameters and increased robustness.  The results clearly underline the improvements of the approach. Considering the fact that the journal reaches a large target audience, the following should be taken into account within the revised manuscript:

- within the abstract, more details in terms of image types, detected classes and the applications that can benefit the proposed transformer should be added[a].

- discussion on what types of application and what image analysis task is implementation suitable for (e.g. Content of Soil petrolium [b])

- moreover, considering the available technologies for computing (GPUs, grid sites, multicores architectures) and the increased amount of hyperspectral image data, it is important for the user to have an estimation for the computational complexity of the algorithm (its components); moreover, processing time measurements on the available hardware would be appreciated. Also, would be important to know which sections of the processing workflow run on GPU and which do not. Moreover, some words on GPU resources management would help if it is the case. Comptational complexity evaluation examples and throughputs and processing time estimations could be found in [c]. Moreover, including discussions on using different approaches for replacing the feature extraction layer using, for example, level set procedures based on PDE approaches as presented in [d], would be of interest.

[a] Signoroni, A.; Savardi, M.; Baronio, A.; Benini, S. Deep Learning Meets Hyperspectral Image Analysis: A Multidisciplinary Review. J. Imaging 20195, 52. https://doi.org/10.3390/jimaging5050052

[b] Shi, P.; Jiang, Q.; Li, Z. Hyperspectral Characteristic Band Selection and Estimation Content of Soil Petroleum Hydrocarbon Based on GARF-PLSR. J. Imaging 20239, 87. https://doi.org/10.3390/jimaging9040087

[c] Baloi, A., Belean, B., Turcu, F. et al. GPU-based similarity metrics computation and machine learning approaches for string similarity evaluation in large datasets. Soft Comput (2023). https://doi.org/10.1007/s00500-023-08687-8.

[d] Baloi, A.; Costea, C.; Gutt, R.; Balacescu, O.; Turcu, F.; Belean, B. Hexagonal-Grid-Layout Image Segmentation Using Shock Filters: Computational Complexity Case Study for Microarray Image Analysis Related to Machine Learning Approaches. Sensors 202323, 2582. https://doi.org/10.3390/s23052582   

The interest for efficient computation and the use of high performance computing resources is also underlined in the following papers:

[1] X. Zhang, Y. Su, L. Gao, L. Bruzzone, X. Gu and Q. Tian, "A Lightweight Transformer Network for Hyperspectral Image Classification," in IEEE Transactions on Geoscience and Remote Sensing, vol. 61, pp. 1-17, 2023, Art no. 5517617, doi: 10.1109/TGRS.2023.3297858.

[2] L. Bruzzone and D. F. Prieto, "A technique for the selection of kernel-function parameters in RBF neural networks for classification of remote-sensing images," in IEEE Transactions on Geoscience and Remote Sensing, vol. 37, no. 2, pp. 1179-1184, March 1999, doi: 10.1109/36.752239.

[3]Yu Bai, Yu Zhao, Yajing Shao, Xinrong Zhang & Xuefeng Yuan. (2022) Deep learning in different remote sensing image categories and applications: status and prospects. International Journal of Remote Sensing 43:5, pages 1800-1847.

[4] Sancho, J.; Sutradhar, P.; Rosa, G.; Chavarrías, M.; Perez-Nuñez, A.; Salvador, R.; Lagares, A.; Juárez, E.; Sanz, C. GoRG: Towards a GPU-Accelerated Multiview Hyperspectral Depth Estimation Tool for Medical Applications. Sensors 2021, 21, 4091. https://doi.org/10.3390/s21124091

[5] Y. Lu, K. Xie, G. Xu, H. Dong, C. Li and T. Li, "MTFC: A Multi-GPU Training Framework for Cube-CNN-Based Hyperspectral Image Classification," in IEEE Transactions on Emerging Topics in Computing, vol. 9, no. 4, pp. 1738-1752, 1 Oct.-Dec. 2021, doi: 10.1109/TETC.2020.3016978.

[6] Caba, J.; Díaz, M.; Barba, J.; Guerra, R.; López, J.A.d.l.T.a. FPGA-Based On-Board Hyperspectral Imaging Compression: Benchmarking Performance and Energy Efficiency against GPU Implementations. Remote Sens. 2020, 12, 3741. https://doi.org/10.3390/rs12223741

Author Response

Thank you for your valuable suggestions. Below are my responses to your comments, one by one.

1.within the abstract, more details in terms of image types, detected classes and the applications that can benefit the proposed transformer should be added[a].

  • The following modifications have been added to the abstract:"Hyperspectral image (HSI) classification is a highly challenging task, particularly in fields like crop yield prediction and agricultural infrastructure detection. These applications often involve complex image types, such as soil, vegetation, water bodies, and urban structures, encompassing a variety of surface features. "
  • The model has been implemented in the aforementioned agricultural monitoring project. The actual detection targets include crops (such as rice, corn, potatoes, etc.) and agricultural infrastructure (houses, canals, ponds, etc.). Considering the confidentiality of agricultural data in practical projects, the original text does not provide a detailed explanation of this aspect. On the other hand, as the project involves self-annotation of data, there is a need to reduce the cost of data labeling. Therefore, the primary goal of this model is to achieve accurate classification with limited data availability.
  • In the first paragraph of Chapter 1 (Introduction), on line 23, add a citation for reference a.

2.discussion on what types of application and what image analysis task is implementation suitable for (e.g. Content of Soil petrolium [b])

  • In the second paragraph of Chapter 4 (Conclusions), on line 422, add the following sentence:"The CESA-MCFormer has demonstrated exceptional versatility in surface object classification tasks. In future research, we intend to further explore the model's application in subsurface exploration tasks (such as soil composition analysis) and are committed to further optimizing its performance."
  • Currently, our research is primarily focused on the classification of surface objects, and we have demonstrated through multiple datasets that the CESA-MCFormer exhibits exceptional versatility in this area. However, we have not yet conducted in-depth research on subsurface objects, such as the soil petroleum content mentioned in your paper. Your suggestion has provided us with significant insights. In our future research, we plan to delve deeper into this field. Thank you very much for your valuable advice.
  • In the first paragraph of Chapter 1 (Introduction), on line 23, add a citation for reference b.

3.moreover, considering the available technologies for computing (GPUs, grid sites, multicores architectures) and the increased amount of hyperspectral image data, it is important for the user to have an estimation for the computational complexity of the algorithm (its components); moreover, processing time measurements on the available hardware would be appreciated. Also, would be important to know which sections of the processing workflow run on GPU and which do not. Moreover, some words on GPU resources management would help if it is the case. Comptational complexity evaluation examples and throughputs and processing time estimations could be found in [c]. Moreover, including discussions on using different approaches for replacing the feature extraction layer using, for example, level set procedures based on PDE approaches as presented in [d], would be of interest.

  • Table 5 has been added, which shows the floating point operations (FLOPS) and the number of parameters for various models, illustrating the computational aspects and complexity of each model.

Reviewer 3 Report

Comments and Suggestions for Authors

(1) It is recommended to include a quantitative evaluation and comparison of the results in the abstract. What are the quantitative results of testing on the three data sets?

(2) In the fourth introduction, it is recommended to add the latest research on Transformer structures in hyperspectral imaging, and not just simply describe and demonstrate the advantages of literature research.

(3) In the fifth introduction, it is recommended to introduce relevant references.

(4) The introduction in paragraphs 6-8 is a bit messy. It is recommended to reorganize it. The advantages and disadvantages of the attention mechanism module in hyperspectral image classification. Why should the attention mechanism be introduced? In the six paragraphs, we only list the attention mechanism modules proposed by different researchers and how these modules are embedded in the model.

(5) in 214-215 lines the input CLS token and HSI features are uniformly divided into 214 eight parts along the spectral feature dimension, each with a feature length of eight why?

(6) The training dataset is a bit small.

(7) It is recommended that true color images and corresponding legends be given in Figures 8 and 9.

(8) It is suggested that the results and discussion be divided into two parts.

Comments on the Quality of English Language

The grammar can be further modified.

Author Response

Thank you for your valuable suggestions. Below are my responses to your comments, one by one.

1.It is recommended to include a quantitative evaluation and comparison of the results in the abstract. What are the quantitative results of testing on the three data sets?

  • The following modifications have been added to the abstract:"To thoroughly evaluate our method, we conducted extensive experiments on the IP, UP, and Chikusei datasets, comparing them with the latest advanced technologies. The experimental results demonstrate that the CESA-MCFormer achieved outstanding performance on all test datasets, with overall accuracies of 96.82%, 98.67%, and 99.59%, respectively."

2.In the fourth introduction, it is recommended to add the latest research on Transformer structures in hyperspectral imaging, and not just simply describe and demonstrate the advantages of literature research.

  • In the fourth introduction (line 50), we added the following text:"In recent studies, researchers continue to explore more lightweight and effective methods for feature fusion and extraction based on the transformer architecture. For instance, Xuming Zhang and others proposed the CLMSA and PLMSA modules, while Shichao Zhang and colleagues introduced the ELS2T."     
  • In the aforementioned statement, I have included two of the latest related studies (from the year 2023). Currently, most research is focused on exploring various feature extraction methods to replace the traditional encoder modules in transformers, as exemplified by the MC module we have employed in this paper.

3.In the fifth introduction, it is recommended to introduce relevant references.

  • In the fifth introduction (line 75), multiple references were added.
  • The comparison results from Tables 5, 6, and Tables 8, 9 somewhat support this argument. Additionally, I have included experimental data in Tables 8 and 9, demonstrating the performance of CESA-MCFormer after applying PCA. The results indicate that the model's predictive accuracy declines after the use of PCA.

4.The introduction in paragraphs 6-8 is a bit messy. It is recommended to reorganize it. The advantages and disadvantages of the attention mechanism module in hyperspectral image classification. Why should the attention mechanism be introduced? In the six paragraphs, we only list the attention mechanism modules proposed by different researchers and how these modules are embedded in the model.

  • In the preceding sections, we discussed research related to hyperspectral image classification. This paper's innovation lies in the integration of attention mechanisms and mathematical morphology, a technique previously introduced into HSI classification tasks by other researchers in various ways. Therefore, I discussed attention mechanisms in paragraphs 6 and 7, and mathematical morphology in paragraph 8. The sixth paragraph cites literature that introduces traditional attention mechanisms into HSI classification tasks without considering the importance of the center pixel. This paper's proposed CESA module considers this importance, which is highlighted in paragraph 7, alongside the similarly focused CAM module. CAM is also one of the main methods compared later in this paper.
  • To improve reader comprehension, I combined paragraphs 6 and 7 into a single section and added a transitional sentence to elucidate the deficiencies in the methods described in the original sixth paragraph.  "However, in HSI classification, a common practice is to segment the HSI into small patches and classify each patch based on its center pixel. Yet, these methods do not sufficiently consider the importance of the center pixel.This approach makes the information provided by the center pixel crucial. "

5.in 214-215 lines the input CLS token and HSI features are uniformly divided into 214 eight parts along the spectral feature dimension, each with a feature length of eight why?

  • The concept of multi-head attention in the Vision Transformer (ViT) (which this paper applies by dividing image features into 8 "heads") allows for parallel processing of different aspects of data, enabling the capture of diverse features and patterns within information more comprehensively. Each "head" focuses on a specific part of the data, allowing the model to analyze information from multiple perspectives, thus enhancing the model's expressive power and depth of understanding. This mechanism is particularly effective in processing complex data, as it can recognize and handle the multi-dimensional complexity of data.

6.The training dataset is a bit small.

  • The model was applied to an agricultural monitoring project, which required self-annotation of data, thus creating a need to reduce data labeling costs. Therefore, the approach proposed in this paper focuses on training with a very small amount of data, as evidenced in the Few-shot Learning Task section at the end of the article.

7.It is recommended that true color images and corresponding legends be given in Figures 8 and 9.

  • We have added the legend as per your request.
  • Due to the numerous channels in hyperspectral images (HSI), there are multiple methods to convert them into RGB three-channel images, each yielding different visualization results and varying in their ability to distinguish different categories. Therefore, choosing a specific visualization method might lead to misunderstandings for the readers. In light of this, we have decided not to include color images in order to avoid potential misguidance, and we apologize for any inconvenience this may cause.

8.It is suggested that the results and discussion be divided into two parts.

  • As per your request, I have revised the last paragraph of Section 3.3, line 398, to follow the format of presenting results first and then discussing them. If there are any more issues that need adjustment, please let me know and I will gladly make further corrections.

Round 2

Reviewer 3 Report

Comments and Suggestions for Authors

(1) It is recommended to use the Kappa evaluation index in the abstract, and at the end you can add a sentence about the promotion and application of the model.

(2) Lines 114 to 116 introduce corresponding references.

(3) In line 155, "along with a learnable matrix of size 65 × 64", is the size correct?

(4) The chart column in Figure 8 needs to be marked with what type each color represents.

(5) The k index of the CESA-MCFormer* model exceeds 98%. How high can it be achieved in practical applications? Have you considered the transferability of the model?

Comments on the Quality of English Language

English proficiency can be further improved.

Author Response

Thank you for your valuable suggestions. Below are my responses to your comments, one by one.

(1) It is recommended to use the Kappa evaluation index in the abstract, and at the end you can add a sentence about the promotion and application of the model.

  • The overall accuracies have been changed to Kappa coefficients.
  • Regarding the promotion and application of the model, the last revision involved two aspects. If you believe further adjustments are still needed, please let me know. Here are the details of the modifications:
  • At the beginning of the abstract:“Hyperspectral image (HSI) classification is a highly challenging task, particularly in fields like crop yield prediction and agricultural infrastructure detection. These applications often involve complex image types, such as soil, vegetation, water bodies, and urban structures, encompassing a variety of surface features.”
  • In the second paragraph of the Conclusions (line 422):"The CESA-MCFormer has demonstrated exceptional versatility in surface object classification tasks. In future research, we intend to further explore the model's application in subsurface exploration tasks (such as soil composition analysis) and are committed to further optimizing its performance."

(2) Lines 114 to 116 introduce corresponding references.

  • A new reference, "Morphological Image Analysis: Principles and Applications," has been added. This publication extensively covers the essential principles and theoretical framework of mathematical morphology and its application across a variety of image processing and analysis tasks.

(3) In line 155, "along with a learnable matrix of size 65 × 64", is the size correct?

  • In the paper, it is correct. As illustrated in Figure 1, combining the CLS Token (64x1) with the feature matrix (64x64) results in a new matrix size of 65x64. To facilitate position encoding, we introduced a learnable matrix of the same size, which is integrated by adding it to the corresponding positions of the original matrix.

(4) The chart column in Figure 8 needs to be marked with what type each color represents.

  • Added the corresponding description.

(5) The k index of the CESA-MCFormer* model exceeds 98%. How high can it be achieved in practical applications? Have you considered the transferability of the model?

  • We conducted experiments on multiple public datasets and achieved excellent results. This to some extent validates the versatility and effectiveness of the model.
  • The performance of the model is closely related to the difficulty of the dataset it is applied to. Currently, the model has been implemented in an agricultural detection project. Given the orderly layout of farmlands and roads, the dataset's difficulty is relatively low, resulting in high classification accuracy.
  • Apart from the agricultural project, this model has not yet been applied in fields outside of public datasets. Therefore, I regret that I cannot provide actual data to prove its effectiveness in other areas.